# Enhanced Open-Hole Strength and Toughness of Sandwich Carbon-Kevlar Woven Composite Laminates

**DOI:** 10.3390/polym15102276

**Published:** 2023-05-11

**Authors:** Mohammad K. A. Khan, Harri Junaedi, Hassan Alshahrani, Ahmed Wagih, Gilles Lubineau, Tamer A. Sebaey

**Affiliations:** 1Department of Mechanical Engineering, College of Engineering, Najran University, Najran 11001, Saudi Arabia; mkkhan@nu.edu.sa (M.K.A.K.); haalshahrani@nu.edu.sa (H.A.); 2Engineering Management Department, College of Engineering, Prince Sultan University, Riyadh 11586, Saudi Arabia; hlukman@psu.edu.sa; 3Mechanical Engineering Program, Physical Science and Engineering Division, King Abdullah University of Science and Technology (KAUST), Thuwal 23955-6900, Saudi Arabia; gilles.lubineau@kaust.edu.sa; 4Mechanics of Composites for Energy and Mobility Lab, King Abdullah University of Science and Technology (KAUST), Thuwal 23955-6900, Saudi Arabia; 5Mechanical Design and Production Department, Faculty of Engineering, Zagazig University, Zagazig 44519, Egypt

**Keywords:** hybrid composites, mechanical properties, fractography, notch sensitivity

## Abstract

Fiber-reinforced plastic composites are sensitive to holes, as they cut the main load-carrying member in the composite (fibers) and they induce out-of-plane stresses. In this study, we demonstrated notch sensitivity enhancement in a hybrid carbon/epoxy (CFRP) composite with a Kevlar core sandwich compared to monotonic CFRP and Kevlar composites. Open-hole tensile samples were cut using waterjet cutting at different width to diameter ratios and tested under tensile loading. We performed an open-hole tension (OHT) test to characterize the notch sensitivity of the composites via the comparison of the open-hole tensile strength and strain as well as the damage propagation (as monitored via CT scan). The results showed that hybrid laminate has lower notch sensitivity than CFRP and KFRP laminates because the strength reduction rate with hole size was lower. Moreover, this laminate showed no reduction in the failure strain by increasing the hole size up to 12 mm. At w/d = 6, the lowest drop in strength showed by the hybrid laminate was 65.4%, followed by the CFRP and KFRP laminates with 63.5% and 56.1%, respectively. For the specific strength, the hybrid laminate showed a 7% and 9% higher value as compared with CFRP and KFRP laminates, respectively. The enhancement in notch sensitivity was due to its progressive damage mode, which was initiated via delamination at the Kevlar–carbon interface, followed by matrix cracking and fiber breakage in the core layers. Finally, matrix cracking and fiber breakage occurred in the CFRP face sheet layers. The specific strength (normalized strength and strain to density) and strain were larger for the hybrid than the CFRP and KFRP laminates due to the lower density of Kevlar fibers and the progressive damage modes which delayed the final failure of the hybrid composite.

## 1. Introduction

Composite laminates are ubiquitous in the aerospace, automotive, construction, and marine industries as structural components [1]. Moreover, the superior mechanical, fatigue, and durability performance of CFRP has supported their application in many primary structures [2,3,4]. Combining various types of materials and components into structures necessitates the existence of holes to facilitate the joining of these distinct components. Thus, it is usual practice and unavoidable in the design, manufacturing, and assembly of structures to introduce holes in composite laminates [5,6]. Hole opening creates a discontinuity in the fiber and the matrix and results in stress concentration during loading. These geometrical perturbations also come with large out-of-plane stresses, while laminates are designed to bear the in-plane load. Holes, as free edges, are systematic hot points in structures, as out-of-plane-stress-related mechanisms, such as delamination, are active in these areas. As a result, studies on opening and using a hole in composite structures have been conducted for many years and are still being conducted for the complicated nature of the damage mechanisms associated with such a problem [7,8,9,10,11,12,13].

The sensitivity of a laminate with the presence of a hole or a notch, in terms of its mechanical properties, is defined by several factors. These factors include the hole or notch dimensions, shape, laminate dimensions, fiber orientations, ply stacking sequence, hole-cutting quality, and materials [14]. Failure stress (strength) and damage propagation vary significantly by changing one or more of these factors, even for unnotched laminates [15,16]. For instance, Wysmulski [17,18] developed FE models and validated them experimentally to study the compression behavior and damage prediction in composite laminates with different cross-sections, demonstrating the sensitivity of these structures to stacking sequences and ply orientations. The most common damage mechanisms associated with the open-hole tensile test are brittle failure, fiber pull-out, delamination, and their combination [14]. Several experimental and numerical investigations have been conducted to determine how the composite laminate responds to a hole. Two types of holes are produced in laminates upon assembly: open hole and filled hole. An open hole is when the hole is left empty, and a filled hole is when the hole is occupied by a bolt or similar. The scenario with the open hole is the topic of this research. Research into the open hole of hybrid composite laminates has been one of the recent focuses [19,20,21,22,23].

Carbon fiber (CF) and Kevlar (aramid) are synthetic fibers known for possessing high strength, stiffness, and low density. The strength and stiffness of CF are higher compared to Kevlar. The superiority of aramid over CF is in its toughness, ability to absorb an impact load, and lower density (1.44 g/cm^3^) [12,13,14,15,16,17,18,19,20,21,22,23,24,25,26,27]. Due to its toughness property, aramid fiber is frequently employed in military applications for ballistic body armor and cut-resistant safety equipment. However, aramid has poor compressive strength, is susceptible to moisture absorption, and is more expensive. As a result, hybrid laminates have been increasingly popular in recent decades due to their coupling properties.

The use of multiple types of fibers in a single laminate allows the designer to fine-tune the properties of the laminate to the design requirements [22,28,29,30,31,32,33,34]. Hybridization is expected to gain some enhancement from some properties. A study by Guled and Chittappa [30] into hybrid CF/Kevlar/Epoxy laminates concluded that using CF laminate on the outer skin enhances the interlaminar shear strength of the overall laminate. Shaari et al. [20,21,23] studied the open-hole tension (OHT) test for Kevlar/glass fiber (GF)/epoxy fiber hybrid laminate composites for different hybridization ratios, hole sizes, and stacking sequences. The OHT of GF laminates improved due to the existence of Kevlar fiber. Hybridization with glass and Kevlar fibers improved hybrid composite specimens’ tensile strength and failure strain. The failure mechanism shifted from being dominated by the matrix to being dominated by the fibers as the hole size increased. De Medeiros et al. [22] investigated the CF/Kevlar hybrid composite laminates. The hybrid laminates that were studied were composed of CF and Kevlar with different orientations. They concluded that the hybrid laminate suffered greater deterioration in mechanical properties when the orientation of the Kevlar fibers was aligned with the direction of the load.

Sebaey and Wagih [26] examined the flexural properties of notched carbon–aramid hybrid laminates in two different forms: intra-ply and sandwich laminates. Sandwich laminates demonstrated progressive damage; meanwhile, catastrophic damage was indicated for intra-ply hybrid laminates. Moreover, sandwich specimens showed higher specific strength compared to intra-ply laminates. Wagih et al. [27] studied the impact and flexural properties of a hybrid carbon–aramid/epoxy composite in the form of a sandwich, in which carbon/epoxy plies were used as face sheets and aramid/epoxy plies were used as the core. They found out that the carbon fiber plies in the bottom part of the laminate were not ruptured after the impact and flexural test. Localized damage was found only at the top carbon fiber ply and the upper plies of the aramid core. Basha et al. [31] examined aramid plies sandwiched between two face sheets of CFRP. CFRP/aramid sandwich composite laminates significantly improved their damage resistance under impact and compression after impact loads and showed better damage tolerance compared to CFRP laminate only.

This study aims to determine how the mechanical performance of composite laminates is affected by the type of fiber and the size of the holes. The composite laminates were constructed out of carbon fiber, aramid fiber, and the hybridization of both. Pristine and open-hole samples were prepared and subjected to a tensile test. The effect of varying hole diameters on mechanical properties was investigated. To understand the failure mechanism, a computed tomography (CT) scan was used to analyze the laminates after failure.

## 2. Materials and Methods

As-received laminated panels used in this study were supplied from Dragonplate, ALLRed & Associates Inc. (Elbridge, NY, USA). Composite laminates with three different types of fiber-reinforced polymers were utilized, namely, carbon-fiber-reinforced polymer, Aramid (Kevlar)-fiber-reinforced polymer, and carbon-Kevlar-fiber-reinforced polymer (hybrid). The hybrid laminate consisted of a single ply of CFRP on the outer layers and Kevlar laminates in the core. The benefits of Kevlar are the higher toughness and resistance to impact. Although not as stiff as carbon fiber, carbon/Kevlar products can be more suitable than one solely comprising carbon fiber or fiberglass in certain applications (bulletproof). Additionally, since Kevlar is non-conductive, it is usually suitable for applications where carbon may cause electromagnetic interference issues, for example, in radomes and UAV components. In all such applications, connecting composite parts requires a hole, for which the notch sensitivity is considered as a key design factor.

After cutting the plies from the woven fiber mat, they were laid, and resin transfer molding was used. The laminate was left to reach its full strength for 20 days. After that, 30 mm was removed from all of the specimen edges to avoid any edge effects. The specimens, with the desired dimensions, and the holes were cut using waterjet cutting.

All of the laminates had a plain woven-roving nature, with 0/90° orientation and a total thickness of 3 mm. The stacking sequences of the three laminates were $[{\left(0/90\right)}_{4}^{C}{]}_{S}$, $[{\left(0/90\right)}_{8}^{K}{]}_{S}$, and $[{\left(0/90\right)}_{2}^{C}/{\left(0/90\right)}_{4}^{K}{]}_{S}$ for CFRP, KFRP, and hybrid, respectively. The CFRP, KFRP, and hybrid densities were 1.45, 1.31, and 1.32 g/cm^3^, respectively. The resin used in this study was NCT 304 with a 140 °C glass transition temperature. Other mechanical properties of the constituent materials are listed in Table 1.

The pristine and OHT test samples were cut from the laminate panels using a water jet machine. The samples were cut to the dimensions of 200 mm × 18 mm for the pristine test samples according to the ASTM D3039M standard [33] and 200 mm × 36 mm according to the ASTM D5766M standard [34] for the OHT test samples. For the pristine samples, tabs of 2 mm thickness made of glass fiber composite with ±45 stacking sequence were glued to the sample edges, leaving a 90 mm gauge length to avoid undesirable damage modes as suggested by the ASTM standard. Meanwhile, the OHT samples were tested without tabs, as the hole presented a higher stress concentration that ensured failure in the hole section. The gauge length of all of the samples, open-hole and pristine, was kept constant at 90 mm. A total of 5 different holes with diameters of 4, 6, 9, 12, and 18 mm were considered to evaluate the notch sensitivity of the produced hybrid sandwich composites, resulting in 5 different width-to-hole-diameter (w/d) ratios of 9, 6, 4, 3, and 2, respectively. Figure 1 shows the samples and X-ray images of the samples, indicating that the samples were free of damages such as delamination and matrix cracking around the hole, which is a critical parameter in evaluating the notch sensitivity of structures.

Tensile tests were performed using an Instron 5882 universal testing machine (Buckinghamshire, UK) with a 100 KN load cell. A mechanical grip was used to clamp samples during loading, as shown in Figure 2. The test was conducted with a crosshead speed of 2 mm/min until there was complete failure for a batch of 3 samples for each configuration. An additional batch of samples with 6 mm hole diameters was tested and interrupted before the final failure to observe the damage progression modes. To avoid confusion, the sample used to characterize the damage mode at a certain loading level was not used again for mechanical characterization.

After testing, the hole area was inspected using an X-ray CT scan (Nikon XT H 225, Nikon Metrology NV, Derby, UK). The prepared samples were then placed inside the scanning machine for examination. The current was set to 140 μA, the voltage was set to 120 kV, the exposure time per frame was 375 ms, and a voxel size of 10.5 μm was set. A total of 2001 projections per scan were collected during the scanning process and the 3D CT Pro reconstruction software was used to reconstruct the scanned images. These images were then sliced and analyzed using Aviso 2.0 software (Redwood City, CA, USA) to inspect the different damage modes inside the samples. So, for CFRP, 3 samples were inspected, where one of them was inspected after final failure while two of them were inspected in different loading stages. For KFRP, a sample was inspected after final failure and another one was inspected before final failure. For the hybrid laminate, four samples were inspected: one after final failure and three in different loading stages.

## 3. Results

Figure 3 shows the sample results of the stress-strain curves for the pristine hybrid laminate and the KFRP laminate with a hole of 6 mm diameter. As is clear from Figure 3, the stress-strain profile shows good repeatability. It is worth remarking that the same repeatability level was monitored for all of the test conditions performed. The elastic regions of the curves are similar and the profiles after the ultimate stress also show similar behavior.

Figure 4 compares the stress-strain curves of different hole diameters for different laminates. The CFRP laminates show a decrease in the ultimate stress and strain at failure by increasing the hole diameter. The stress-strain profiles for different hole diameters and pristine samples are similar. The stress rises linearly until reaching the maximum stress, and then a progressive decrease in the stress occurs, followed by a complete loss of its resistance. Unlike CFRP, KFRP laminates show a standard brittle behavior for the pristine and OHT samples, where the stress increases linearly with increasing strain until a sudden stress drop occurs, causing the complete loss of the load-bearing capacity of the laminate, indicating brittle failure. The hybrid laminates show a different profile between the pristine and OHT samples. The pristine hybrid sample shows a typical brittle stress–strain curve similar to the KFRP laminates. However, the OHT samples show progressive failure, indicating the failure of different plies of fibers (Kevlar plies and CF plies). Figure 5 shows the typical macroscopic failure modes in the samples with different hole sizes. It demonstrates that the KFRP laminate suffered from brittle localized fiber breakage in the high stress concentration area. On the other side, CFRP and hybrid laminates show dispersed damage modes around the hole area, which explains the progressive damage mode observed in the stress-strain behavior (see Figure 4a,c). Table 2 summarizes the modulus and the strength and strain at failure for the three configurations with different hole diameters. It is worth noting that the modulus here is calculated based on the crosshead displacement, as in most of the samples, the extensometer values are not available.

## 4. Discussion

### 4.1. Damage Sequence

Figure 6 displays the CT scan image of samples with a 6 mm diameter hole after failure. It justifies the different apparent failure modes experienced by each sample type. Delamination dominates the CFRP’s failure mode, as seen in Figure 6a. Fiber breakage is also observed near the hole. For KFRP in Figure 6b, the failure mode is governed by fiber breakage, followed by cracks propagating throughout the matrix between the layers and repeating until fully broken. Delaminations are completely inhibited for KFRP laminates due to the fact that Kevlar is much less anisotropic compared to carbon fiber. Therefore, the generated out-of-plane stresses are less. For the hybrid laminate, fiber breakages of Kevlar and the separation of the CF laminates from the Kevlar laminate are observed in Figure 6c.

To better understand the different damage modes, the evolution of these damage modes, and how they interact with each other, some tests were interrupted before the final failure of the laminate and the damage modes were examined. Figure 7 shows the damage sequence of the CFRP laminate with a hole diameter of 6 mm in different stages of loading, before the laminates ultimately failed, and the corresponding stress-strain curves. The damage modes are shown using a 2D cross-sectional view of the laminate in 2 perpendicular planes at the hole center (Figure 7c) and at the edge of the hole (Figure 7d), as explained in the 3D micrograph in Figure 7b. Matrix cracks appeared first around the hole edge early during loading, where no significant change in the stress–strain profile could be observed. These matrix cracks initiated inside the tows are marked with yellow in Figure 7c,d due to the high interlaminar stress concentrations on the matrix and the free edge effect around the hole. The presence of the hole and the free edge induced out-of-plane stress that generated interlaminar shear stress [35]. Just before the final failure, “Point 2” in Figure 7a, delaminations were observed (see Figure 7c) that were initiated at the hole edge due to the interlaminar shear stress generated near the hole [36]. Similar phenomena have been reported elsewhere [14,15,37,38]. These delaminations were initiated and propagated around the hole in both directions, as shown in Figure 7d, due to the high shear stresses generated and the woven-roving nature of the plies. The propagation of delaminations reflects small stress drops in the stress–strain curve just before “Point 2”. These stress drops were repeated before the final failure due to the propagation of more delaminations until the saturation of the composite with delamination. Once the laminate was saturated with delaminations, fiber kinking and a fiber cut occurred, as shown in Figure 7c,d, causing a large stress drop that reduced the laminate strength to almost zero, referred to as “Point 3”. In this stage, due to the unstable growth of damage, extensive matrix cracks and delaminations propagated throughout the whole laminate width, as shown in Figure 7d.

Figure 8 shows the damage sequence in KFRP with a 6 mm hole diameter. The damage modes were recorded in two stages: during the load, and before and after the maximum stress (Point 1 and Point 2). As previously stated, the stress-strain response is monotonically linear until the final brittle failure. It is demonstrated that, in the loading direction, the hole edges do not suffer from any kind of damage even after the final failure, as shown in Figure 8c. The damage was very localized in the hole center in the perpendicular direction to the loading. At “Point 1”, single delamination of the first layer was observed in the direction perpendicular to the loading direction (see Figure 8d), which can be justified by the cut of fibers during the machining of the holes that produced a larger stress concentration at this particular interface and hence delamination growth. Apart from this single delamination, unlike the CFRP laminate, there was no sign of any kind of damage throughout the laminate thickness. It is well known that KFRP laminates possess lower interlaminar shear stress (ILSS) compared to CF/epoxy composite laminates [28]. In contrast to CFRP, no delamination was shown on KFRP, even though matrix cracking throughout the thickness was observed. This can be justified by the lower tensile strength (lower stress level) of KFRP compared to CFRP, as shown in Figure 4. The damage mode, after failure, indicated at “Point 2” in Figure 8a, shows a brittle cut of fibers throughout the whole laminate thickness with few pulled out fibers. During the formation of the fiber bundle pull-out, a few local delaminations appeared just around the hole. In summary, the damage mode in KFRP was different than that of CFRP. In KFRP, localized damage in the hole center occurred, whereas CFRP showed distributed damage throughout the whole width of the sample with extensive matrix cracks and delaminations.

The damage sequence in the hybrid laminate with a 6 mm hole diameter is shown in Figure 9. In this laminate, we tested 4 samples under different loading strains until complete failure, defined as “Points 1, 2, 3, and 4” in Figure 9a. The stress increased monotonically by increasing the strain until “Point 1”, where local delamination at the CFRP/Kevlar interface was initiated around the hole (see Figure 9c,d) due to the low interlaminar toughness of this interface [26]. Owing to the free edges at the hole, out-of-plane stresses acted as opening stresses at the interface between the CFRP and Kevlar layers. Once these stresses reached the interlaminar shear strength of that interface, delaminations were initiated. These delaminations propagated until “Point 2”, as shown in Figure 9c,d. At “Point 2”, the initiation of the Kevlar fiber cut occurred at the lower Kevlar plies (see Figure 9(d2)) due to the loss of the support from the neighbor CFRP plies, as a result of the separation between the CFRP and Kevlar layers. Just after “Point 2”, a large stress drop occurred due to the complete fiber cut in the Kevlar plies’ core, as shown in “Point 3”. Additionally, extensive delaminations at the CFRP/Kevlar interface occurred in this stage, as shown in Figure 9(c3). It is notable that two interesting phenomena happened with the KFRP core in the hybrid laminate that did not appear in the KFRP laminate. The first is that Kevlar fiber breakage (cut) occurred progressively due to the support of the Kevlar plies with CFRP plies, which increased their in situ strength. The second interesting phenomenon is that the failure strain of the Kevlar core was larger than the failure strain of the pure Kevlar laminate. For the pure Kevlar laminate, the failure strain was 0.046, whereas the initiation failure strain of the Kevlar core in the hybrid laminate was somewhere between “Points 1 and 2”, corresponding to strains 0.06 and 0.072. Moreover, the strength of the Kevlar core plies in the hybrid laminate was increased compared to the pure Kevlar plies in the KFR laminate due to the in situ strength effect. Therefore, the CFRP face plies strengthened and toughened the Kevlar plies’ core. After “Point 3”, the CFRP face sheets were the main load carrier during loading, which supported the material until the failure strain of CFRP was reached and then complete failure occurred, see Figure 9(c4,d4). The failure strain of the CFRP plies in the hybrid laminate was equal to that of the pure carbon fiber laminate. Moreover, extensive matrix cracks in the CFRP and Kevlar core plies were observed after failure, as shown in Figure 9(c4,d4).

Figure 10 shows the damage modes in samples with a 12 mm hole diameter to evaluate the effect of hole size on the damage modes. As shown in the figure, the damage modes observed after failure were similar to those observed for laminates with a 6 mm hole diameter, as shown in Figure 7, Figure 8 and Figure 9. Therefore, we can summarize the damage modes in each of the three configurations considered. For the CFRP samples, progressive damage was observed as matrix cracking initiation and propagation occurred, followed by delamination at the free edges around the hole. Finally, fiber breakage and kinking at the hole edges and extensive delamination throughout the whole laminate width occurred. For the KFRP samples, the damage mode was a pure brittle failure with fiber cut and pull-out. For the hybrid laminate, a progressive damage mode was observed as delamination (separation) was initiated at the CFRP/Kevlar interface and then propagated, which weakened the Kevlar plies in the laminate core. After that, the initiation of the fiber cut in the Kevlar plies in the laminate was followed by the propagation of the fiber cut in the Kevlar plies. Finally, the fiber was cut in the CFRP plies, which resulted in complete failure.

### 4.2. Notch Sensitivity

Figure 11a shows the tensile strength of the three configuration composites as a function of the hole diameter. The strength of the laminates decreased with the increase in the open-hole diameter of all types of laminates. A hole in a laminate produced stress concentration and out-of-plane stresses. These two factors altered the failure mechanisms and degraded the strength of the laminate, which was well established for the OHT test [39]. Other studies also confirmed that larger holes caused a decrement in the tensile strength of the composites. Stress could be disseminated to a larger region of the specimen cross-section with a small hole diameter, whereas with a larger hole, the damage zone extended to a greater portion of the specimen; therefore, less of a region remained to support the stress [15,21].

Figure 11b shows the evolution of the strain upon failure with an increase in different hole diameters. Similar to the tensile strength, strain at failure also showed a drop due to the increase in the open-hole diameter, with an exception being the hybrid laminate. The higher drop in the strain at failure was experienced by the KFRP composites. The drop in the strain due to the increase in the hole diameter is due to the stress concentration at the hole, which leads to the early failure of laminates. Meanwhile, a slight increase in the strain at failure was observed for hybrid laminates with 4, 6, and 8 mm hole diameters. The increase in strain at failure for the hybrid OHT sample, as seen from the failure mechanics (Figure 11b) and stress–strain curves (Figure 4), can be justified by the delamination of the CF plies. As seen in Figure 9, hybrid laminates showed two steps of failure. The first one occurred at the maximum stress due to the breakage of the Kevlar fiber plies, whereas the second one was due to the failure caused by the delamination between the CF plies and the Kevlar plies. The enhancement in the ductility of the OHT hybrid laminate samples could be one of the advantages of the laminate compared to the other two laminates.

Normalized strength was calculated by dividing the open-hole tensile strength of each composite system by the tensile strength of the pristine composite. The plot of normalized strength to the w/d is presented in Figure 12a. For a hole diameter of 4 mm (w/d = 9), a drop to 72.1% of the pristine sample for samples was observed for CFRP, whereas, for the same ratio, 69.5% and 61.5% were observed for the hybrid and KFRP laminates, respectively. Meanwhile, at w/d = 6, the lowest drop in strength was shown by the hybrid laminate at 65.4% of the pristine sample, followed by CFRP and KFRP with 63.5% and 56.1%, respectively. At the largest hole diameter of 18 mm (w/d = 2), the lowest strength drop was shown by CFRP with 39.7%, followed by the hybrid and KFRP laminates with 33.9% and 29.4%, respectively. In summary, the KFRP composite had the highest drop for all conditions compared to CFRP and the hybrid laminate. On the other hand, the hybrid composite laminate normalized OHT strength was very close to that of CFRP. A drop in the OHT strength for all of the samples was observed by the decrease in w/d. The same trends have also been shown by Salleh et al. [19] for fiberglass-reinforced epoxy composites. KFRP is more notch-sensitive compared to CFRP. The replacement of the outer layers of KFRP with face CFRP plies to produce the hybrid laminate reduces the notch sensitivity of the KFRP laminate.

The normalized strain was also calculated, and the result is presented in Figure 12b. The figure displays the normalized strain of the three laminates and the change in the w/d. It shows that KFRP had the highest drop in the strain at failure in comparison to its pristine sample. The hybrid laminates had higher normalized strain values compared to CFRP. The presence of CFRP plies in the outer layers of the KFRP laminates enhanced the OHT strength and the laminate’s ductility simultaneously.

In comparison to CF, Kevlar has a lower density, which also results in composite laminates that have lower density. Thus, normalized strength and strain in regard to density (specific strength and strain, respectively) were also calculated, and the result is presented in Figure 13. It shows that hybrid laminate has higher normalized strength/density (Figure 13a) and normalized strain/density (Figure 13b) compared to CFRP and KFRP for the OHT samples. It showed almost around 7% and 9% larger specific strength than the CFRP and KFRP laminates. More interestingly, it showed a large enhancement in specific strain at a larger width-to-hole ratio reaching 20% and 18% compared to CFRP and KFRP laminates.

## 5. Conclusions

Pristine and open-hole tension (OHT) composite laminates of pure CFRP and KFRP and a hybrid laminate with a core of KFRP surrounded by two face sheets of CFRP were tested. Micro-computed tomography was deployed to characterize the damage modes and their sequence during the tensile test. Based on the results and discussion, the following conclusions can be drawn:The progressive damage mode occurred for CFRP and the hybrid laminate under the OHT test, while KFRP showed a brittle damage mode.For CFRP, the damage was initiated as matrix cracks around the hole due to the interlaminar shear stresses at the hole edges, followed by delaminations at the same position. Then, fiber breakage occurred progressively after the saturation of the laminate with delaminations. For the hybrid laminate, the damage was initiated as delaminations at the CFRP/Kevlar interface that propagated by increasing the applied strain. Then, the Kevlar core fiber cut was initiated and propagated, causing a partial loss of laminate strength. Finally, CFRP face sheet fiber breakage and matrix cracking occurred, causing the final failure of the samples.The hybrid laminate showed better notch sensitivity than the CFRP and KFRP laminates due to the progressive damage mode, where the reduction in the strength of the hybrid laminate by increasing the hole size was the lowest.The hybrid laminate showed 7 and 9% higher specific strength compared to the CFRP and KFRP laminates. At the same time, it showed a 20% and 18% higher specific failure strain compared to the CFRP and KFRP laminates.

## Figures and Tables

**Figure 1 polymers-15-02276-f001:**
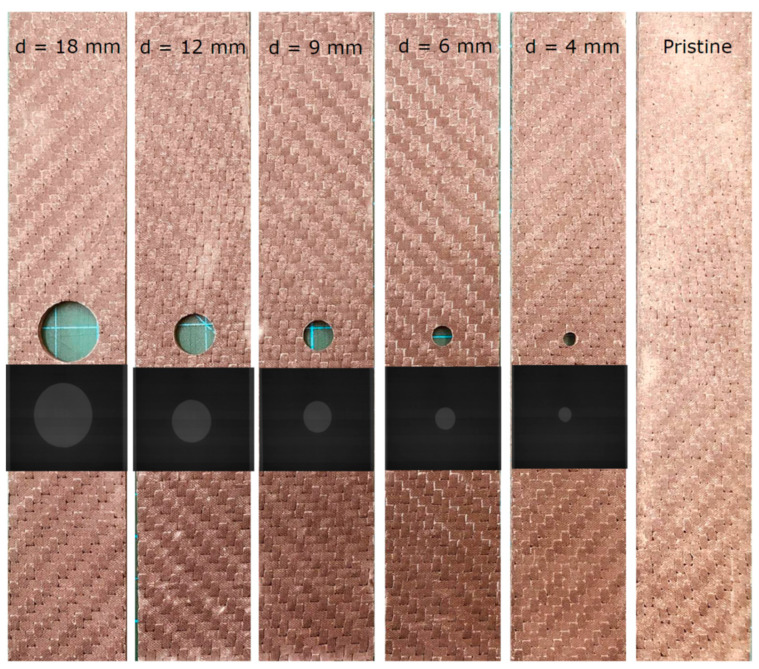
Photograph of the samples with different hole diameters and X-ray images around the hole.

**Figure 2 polymers-15-02276-f002:**
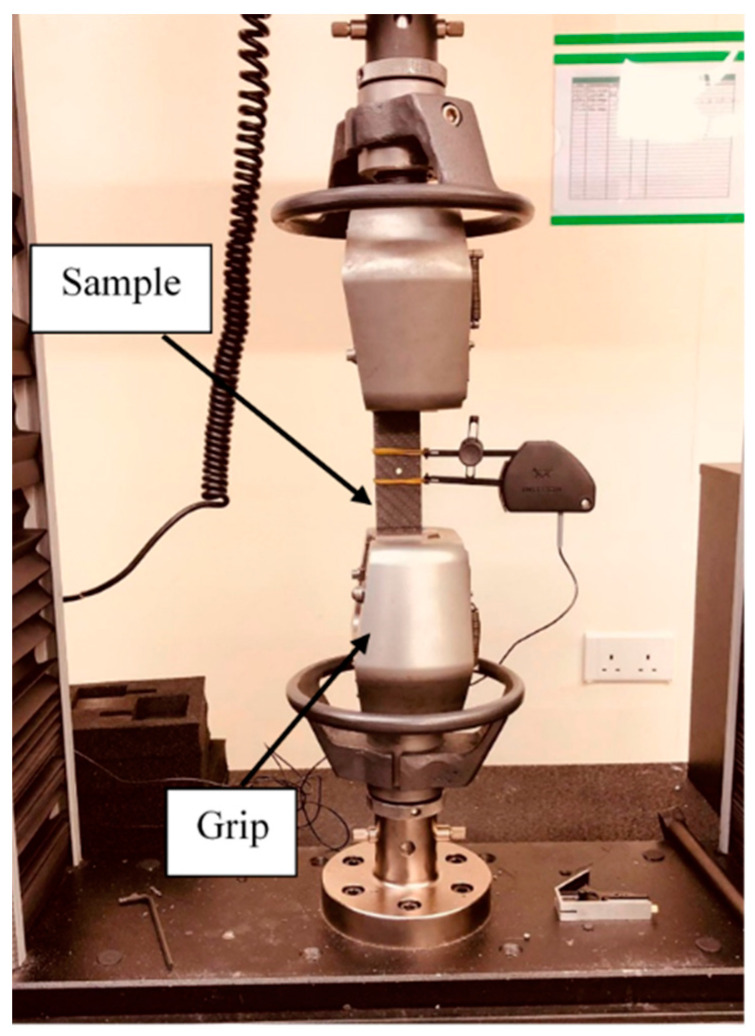
Experimental tensile test setup.

**Figure 3 polymers-15-02276-f003:**
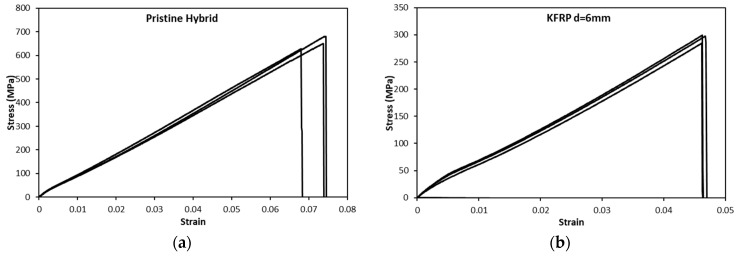
Stress-strain curve repeatability of (**a**) pristine hybrid and (**b**) KFRP open-hole composite laminates, d = 6 mm.

**Figure 4 polymers-15-02276-f004:**
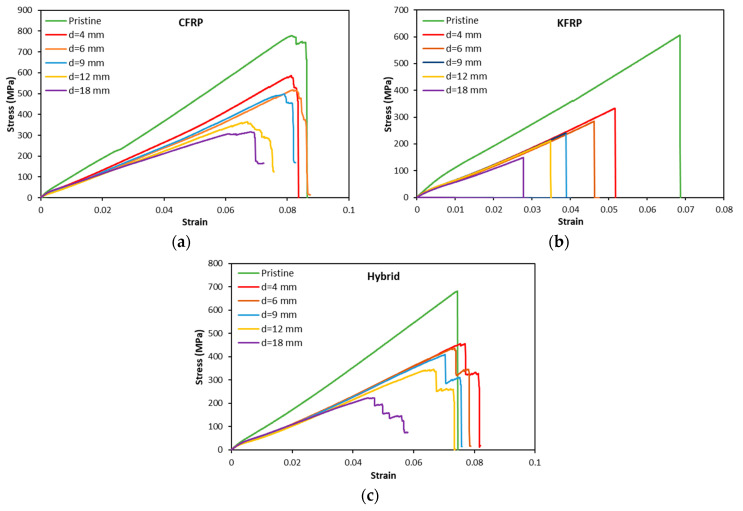
Stress-strain curves of OHT samples of different hole diameters for (**a**) CFRP, (**b**) KFRP, and (**c**) hybrid.

**Figure 5 polymers-15-02276-f005:**
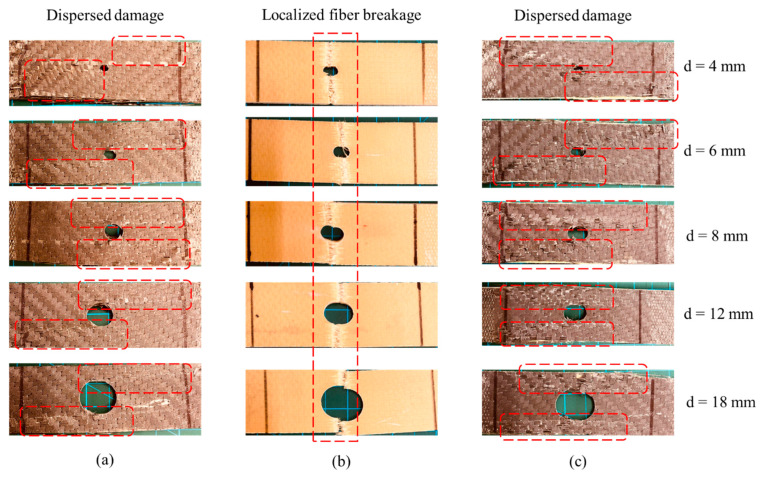
Open-hole samples after the tensile test: (**a**) CFRP, (**b**) KFRP, and (**c**) hybrid. (Red dotted frames show the damage area).

**Figure 6 polymers-15-02276-f006:**
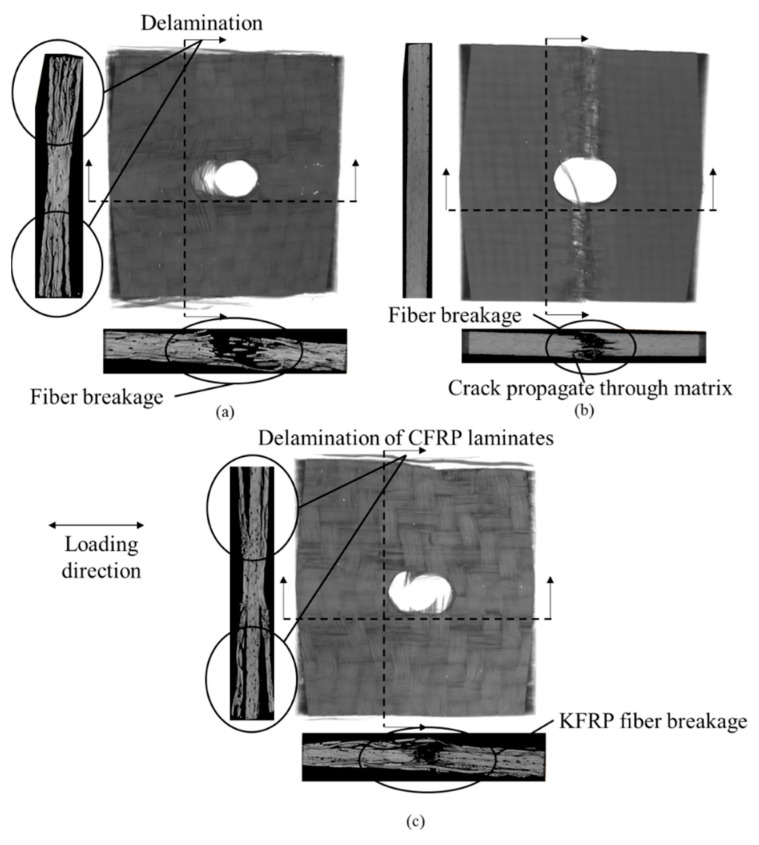
CT scan images of failed open-hole tension samples for 6 mm hole diameter: (**a**) CFRP, (**b**) KFRP, and (**c**) hybrid.

**Figure 7 polymers-15-02276-f007:**
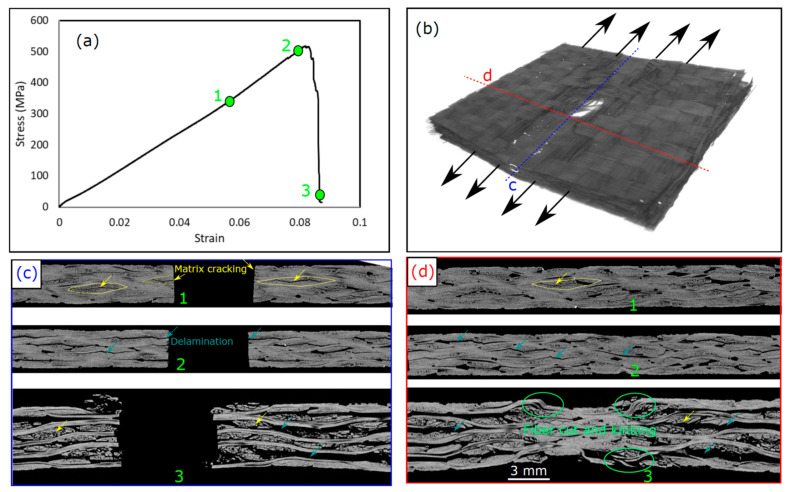
Damage sequence in CFRP OHT laminate with 6 mm hole diameter: (**a**) stress-strain curve, (**b**) 3D CT-micrograph of the damage after failure at “Point 3”, (**c**) 2D cross-section at “Points 1, 2, and 3” in the loading direction at the sections marked in (**b**), (**d**) 2D cross-section at “Points 1, 2, and 3” in the perpendicular to the loading direction at the sections marked in (**b**).

**Figure 8 polymers-15-02276-f008:**
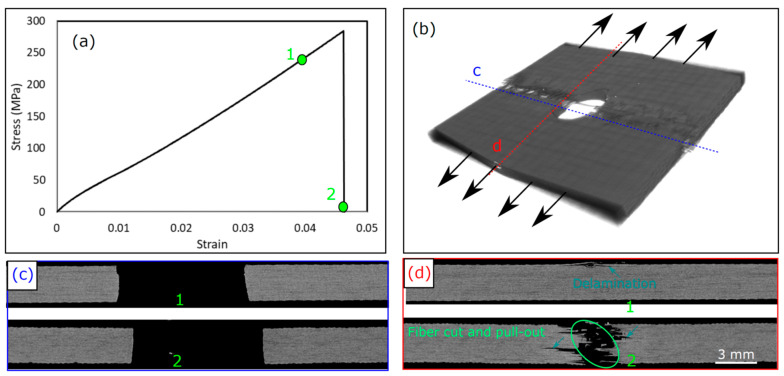
Damage sequence in KFRP OHT laminate with 6 mm hole diameter: (**a**) stress–strain curve, (**b**) 3D CT-micrograph of the damage after failure at “Point 2”, (**c**) 2D cross-section at “Points 1 and 2” in the loading direction at the sections marked in (**b**), (**d**) 2D cross-section at “Points 1 and 2” in the perpendicular to the loading direction at the sections marked in (**b**).

**Figure 9 polymers-15-02276-f009:**
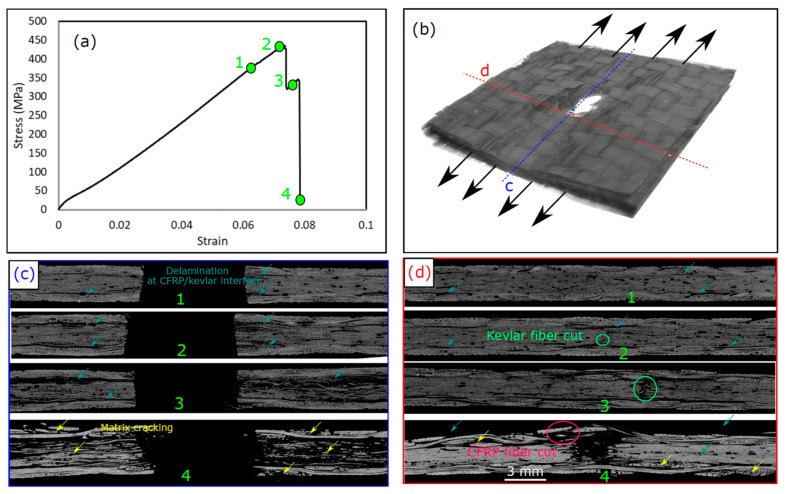
Damage sequence in hybrid OHT laminate with 6 mm hole diameter: (**a**) stress-strain curve, (**b**) 3D CT-micrograph of the damage after failure at “Point 4”, (**c**) 2D cross-section at “Points 1, 2, 3 and 4” in the loading direction at the sections marked in (**b**), (**d**) 2D cross-section at “Points 1, 2, 3 and 4” in the perpendicular to the loading direction at the sections marked in (**b**).

**Figure 10 polymers-15-02276-f010:**
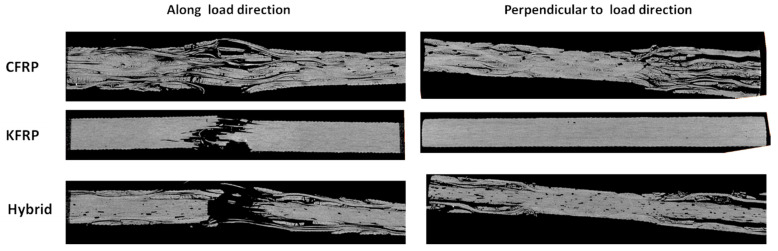
CT scan cross-section of failed open-hole tension samples with 12 mm hole diameters near the hole.

**Figure 11 polymers-15-02276-f011:**
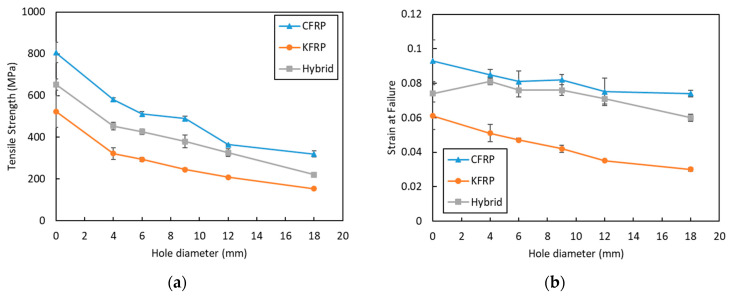
(**a**) Tensile strength and (**b**) strain upon failure of the composite laminates with different hole diameters.

**Figure 12 polymers-15-02276-f012:**
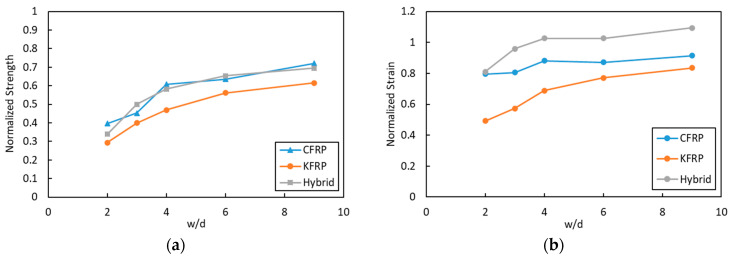
(**a**) Normalized strength and (**b**) normalized strain at different w/d.

**Figure 13 polymers-15-02276-f013:**
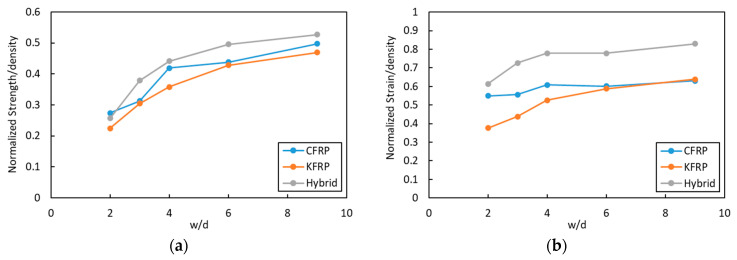
(**a**) Normalized strength/density and (**b**) normalized strain/density at different w/d.

**Table 1 polymers-15-02276-t001:** Mechanical properties of the fiber and matrix used.

	Carbon Fiber	Kevlar Fiber	Epoxy Matrix
Tensile strength (MPa)	3500	2920	66
Tensile modulus (GPa)	230	70.5	3.7

**Table 2 polymers-15-02276-t002:** Strain and stress at failure for the test campaign.

Hole Diameter (mm)	Apparent Modulus (MPa)			Strain at Failure			Tensile Strength (MPa)		
	CFRP	KFRP	Hybrid	CFRP	KFRP	Hybrid	CFRP	KFRP	Hybrid
0	9482 ± 220	8733 ± 399	8786 ± 369	0.093 ± 0.012	0.061 ± 0.008	0.074 ± 0.005	806 ± 48	524 ± 76	653 ± 27
4	6426 ± 139	6183 ± 109	5872 ± 171	0.085 ± 0.003	0.051 ± 0.005	0.081 ± 0.002	581 ± 8	322 ± 28	454 ± 19
6	6458 ± 093	6122 ± 215	5712 ± 489	0.081 ± 0.006	0.047 ± 0.001	0.076 ± 0.004	512 ± 11	294 ± 9	427 ± 14
9	6030 ± 347	6112 ± 302	5693 ± 162	0.082 ± 0.003	0.042 ± 0.002	0.076 ± 0.003	490 ± 12	246 ± 6	380 ± 31
12	5748 ± 562	5851 ± 334	5359 ± 078	0.075 ± 0.008	0.035 ± 0	0.071 ± 0.003	365 ± 2	209 ± 4	326 ± 18
18	5372 ± 011	5284 ± 331	5149 ± 392	0.074 ± 0.002	0.03 ± 0.001	0.06 ± 0.002	320 ± 14	154 ± 4	221 ± 9

## Data Availability

The data will be made available upon request to the authors.

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
