# Peer review of "Enhanced Open-Hole Strength and Toughness of Sandwich Carbon-Kevlar Woven Composite Laminates"

_polymers, 2023, doi:10.3390/polym15102276_

Round 1

Reviewer 1 Report

This paper investigates the notch sensitivity enhancement of hybrid carbon/epoxy (CFRP) composite with Kevlar core sandwich compared to monotonic CFRP and Kevlar composites. The results showed that hybrid laminate has lower notch sensitivity than CFRP and KFRP laminates because the strength reduction rate with hole size was lower. Some research results obtained are meaningful to promote the applications of open holes composites in engineering structures. However, the authors are encouraged to consider the following minor comments for necessary improvement.

1.      Abstract:

(1) For the improvement of mechanical properties of composite after fiber hybrid, please provide specific quantitative indicators.

(2) The authors are advised to enrich the discussion of research methods, and then condensed the research results.

2.      Introduction:

(1) The authors said “Composite laminates are ubiquitous in the aerospace, automotive, construction, and marine industries as structural components”, the main reasons were the superior mechanical, fatigue and durability performances of FRP. The reviewer suggested that the authors add the relevant research background to support the above viewpoint in the introduction. The following relevant studies can be reviewed, such as “Mechanics of Advanced Materials and Structures, 2023, 30(4):814-834.”, “Composite Structures. 2021, 261: 113285.”, “Engineering Structures, 2023, 274: 115176.”.

(2) Please delete irrelevant paragraphs of “The introduction should briefly place the study in a broad context and highlight why it is important. It should define the purpose of the work and its significance. The current state of the research field should be carefully reviewed and key publications cited. Please highlight controversial and diverging hypotheses when necessary. Finally, briefly mention the main aim of the work and highlight the principal conclusions. As far as possible, please keep the introduction comprehensible to scientists outside your particular field of research. References should be numbered in order of appearance and indicated by a numeral or numerals in square brackets—e.g., [1] or [2,3], or [4–6]. See the end of the document for further details on references.”

3.      Materials and Methods:

(1) The mechanical properties of original materials should be listed in the table.

(2) The OHT samples were tested without tabs, how to ensure that clamping has no effect on the sample?

4.      Results and discussions:

(1) Please provide the tensile modulus of open hole samples in table 1.

(2) Please analyze whether the tensile strength and modulus of the hybrid specimens meet the mixing ratio.

(3) Delete the words of results in Page 12, and do full text check for similar errors.

5.      Conclusion:

The conclusion is suggested to be further condensed according to the important findings, give the key results and innovative findings by point to point.

Round 2

Reviewer 1 Report

Accept.

Reviewer 2 Report

Accept in present form